# The Servqual Method as an Assessment Tool of the Quality of Medical Services in Selected Asian Countries

**DOI:** 10.3390/ijerph19137831

**Published:** 2022-06-26

**Authors:** Aleksandra Jonkisz, Piotr Karniej, Dorota Krasowska

**Affiliations:** 1Department of Dermatology, Venerology and Pediatric Dermatology, Medical University of Lublin, 20-081 Lublin, Poland; dor.krasowska@gmail.com; 2Faculty of Finances and Managment, WSB University in Wroclaw, 53-609 Wroclaw, Poland; piotr@karniej.pl

**Keywords:** servqual, servqual meta-analysis, patient satisfaction, quality of medical services

## Abstract

Introduction: The Servqual (an acronym from the words “service” and “quality”) method is used to assess the quality of provided services on the basis of standardised evaluation parameters. This method is based on five gaps resulting from the discrepancy between expected and received service quality. The aim of this meta-analysis and the systematic review was to view and assess the major differences in the five dimensions of the Servqual method used to evaluate the quality of delivered health care services in selected Asian countries. Another goal of the study was to confirm the use of the Servqual method as a suitable tool for assessing the quality of health care services. Methods: This study followed the PRISMA guidelines for systemic reviews and meta-analyses. The following electronic databases for medical publications were used: PubMed, Medline, Scopus, and Cochrane were searched for articles published from January 2000 to April 2020. The databases were explored with original search queries containing the following terms: “Servqual”, “service quality”, “Servqual model”, “servqual questionnaire”, “health service quality”, “health care services”, “patients’ expectation”, “patients’ perception”, “expectation”, “perception”, and “health care services”, in combination using “AND” and “OR”. In order to minimize bias, two researchers (PK and DK) independently performed an online search for peer-reviewed papers, using the combinations of the above-mentioned words. In addition, references of eligible publications were checked. All disagreements, regarding the inclusion or exclusion of specific studies, were resolved through consultations among all the authors. Results: A total of 96 reports were identified and submitted to a preliminary screening selection. As a result of the pre-screening stage, 64 papers were qualified to further evaluation. The output of the evaluation brought 15 reported studies, meeting the pre-defined inclusion/exclusion criteria. The total number of participants was 5903 (ranging from 20 to 439 in individual reports), and 54% of them were women. Eight studies (53%) were from Iran, two from Pakistan (13%) and one each from Arabia, Malaysia, South Korea, Bangladesh, and Iraq (each-about 7%). The results showed gaps between patients’ expectations and perceptions in all five dimensions of Servqual in almost each analysed study. The highest and lowest values of the gaps in quality scores were associated with the dimensions of reliability, tangibility, empathy, and responsiveness, respectively. Conclusions: The study demonstrated that the method of Servqual is broadly used in various medical sectors to assess the quality of medical services provided. In addition, the study demonstrated that patients had significantly higher expectations of the medical services offered in the five dimensions studied. The results, obtained through the Servqual method, may help improve and monitor the quality of services provided by different institutions.

## 1. Introduction

High quality levels of medical care for patients are an overarching goal of all healthcare systems throughout the entire world. Already in the 1980s, determining the quality of services and products became a key objective for service providers. Service quality has many dimensions, thus its assessment is really challenging. Parasurmman et al. perceived it as imperative that most objective quality measures were introduced in all provided services, including medical ones [1]. On the basis of their own observations and research, they designed and developed a Servqual questionnaire in which discrepancies between customer expectations and the service provided were identified by analysing five gaps/dimensions. The quality of provided services is assessed through the dimensions of tangibility, assurance, empathy, responsiveness, and reliability. The size of this gap/dimension exerts a relatively significant impact on the final quality of the service [1]. According to Parasurman et al., it is the client/patient who determines the quality of the service, which is a product of their perceptions and expectations [2]. The implementation of the Servqual questionnaire has given the managers of healthcare units a tool, enabling them to identify the weakest links in the hospital/clinic systems but, above all, to strengthen the systems and to closely monitor their functions, which, as a whole, is an essential component of the quality improvement process of the services provided [3]. Healthcare units, identifying gaps in provided service, submit them to objective and detailed analyses to implement, at the end, appropriate compensatory/corrective actions [4]. The World Health Organisation (WHO) recommends the use of patient satisfaction scores in treatment programmes to guide efforts to improve the quality of provided healthcare. Naqavi et al. showed significant relationships among patient satisfaction, the therapy process, and the maintenance of treatment outcomes [5]. Practically all over the world, the managers of medical institutions are expected to raise the level of quality of medical services provided to patients by ensuring the highest possible quality of care in the entities they manage [4]. Nadi et al. noted that the problem of inadequate quality of medical service affected mainly those facilities that were not focusing on understanding and meeting patients’ needs and requirements. According to the authors, the managers of healthcare units should identify with their customers’ priorities, setting up their own policy models in consideration of client/patient feedback. The lack of a direct relationship with customers/patients is a serious obstacle to learning and meeting their expectations [6].

One of the inclusion criteria was the location of the study subject in Asia, due to a very rich representation of reports from this region of the world. The existing source literature may lead to a conclusion that the SQ method is very often used to improve the quality of medical services in Asia. This is why the authors decided to subject this region to a more detailed survey.

Goal of the study The aim of the study was a systematic review of the literature in terms of assessing the quality of provided medical services using the Servqual method.

## 2. Materials and Methods

This study followed the PRISMA guidelines for systemic reviews and meta-analyses [7]. The following electronic databases for medical publications were used: PubMed, Medline, Scopus, and Cochrane were searched for articles published from January 2000 to April 2020. The databases were explored with original search queries containing the following terms: “Servqual”, “service quality”, “Servqual model”, “servqual questionnaire”, “health service quality”, “health care services”, “patients’ expectation”, “patients’ perception”, “expectation”, “perception”, and “health care services”, in combination using “AND” and “OR”. In order to minimize bias, two researchers (PK and DK) independently performed an online search for peer-reviewed papers, using the combinations of the above-mentioned words. In addition, references of eligible publications were checked. All disagreements, regarding the inclusion or exclusion of specific studies, were resolved through consultations among all the authors.

### 2.1. Eligibility Criteria

The inclusion criteria were as follows: (1) patient population over 15 years of age; (2) the place of provided services: public hospitals, private hospitals, and outpatient clinics; (3) studies must be complete, peer-reviewed, reported in English, and use the Servqual questionnaire as an assessment tool; (4) research must be carried out in Asian countries; (5) the research must contain all the necessary data from the Servqual questionnaire, necessary for the meta-analysis; and (6) the questionnaire of SQ must include a minimum of 20 questions divided for a minimum of 5 dimensions (tangibility, responsiveness, empathy, assurance, and reliability) and use Likert’s scale 0–5 (where 0 means “not agree” and 5 means “definitely agree”).

The exclusion criteria: (1) research carried out on persons: minors, students, and war veterans; (2) patient satisfaction research with the use of medical equipment; (3) research from continents other than Asia; (4) studies that did not provide necessary data from the Servqual questionnaire to perform meta-analysis; (5) all non-English researches; and (6) other non- 20 questions SQ models.

### 2.2. Statistical Analysis

Quality of health care services was statistically analysed in five dimensions. Mean ± standard deviation values were provided in all studies for patients’ perceptions and expectations. The mean difference was used as the effect size for all health care services in the studies included in our analysis. Cochran’s (Q) heterogeneity statistic (*p* < 0.1) and the I-squared test were used to evaluate the magnitude of heterogeneity among the true effect sizes. A forest plot was generated with the results for mean difference (MD) values with 95% assurance intervals (95% CI). Subgroup analysis and meta-regression were used to evaluate associations between the calculated mean difference and the characteristics, such as the year of publication, gender, age, marital status, and the location (country) of the study. A potential publication bias was explored with Egger’s test, and the trim and fill technique was used to adjust the pooled estimates for the likelihood of missing studies. The statistical analysis was performed using the Statistica 13 software (TIBCO Software Inc., Palo Alto, CA 94304, USA).

## 3. Results

A total of 96 studies were identified and subjected to the initial screening. We then pre-selected 64 articles for further full-text reviews. That step finally revealed 15 studies [4,5,6,8,9,10,11,12,13,14,15,16,17,18,19] that were found to meet our inclusion criteria (Figure 1).

A total of 15 studies were included in the metanalysis. The total number of participants was 5903 (ranging from 20 to 439 in individual reports), and 54% of them were women. The majority (83%) of the participants were married. The mean age of the whole group was 38 years. Eight studies (53%) were from Iran, two from Pakistan (13%) and one each from Saudi Arabia, Malaysia, South Korea, Bangladesh, and Iraq (each–around 7%) (see Table 1). Eleven studies dealt with assessments at hospitals, four at healthcare centres and one each at: a public clinic, a treatment centre, and a haemodialysis centre (see Table 1).

### Dimensions

**Tangibility and empathy**—the largest gaps between patients’ expectations and perceptions were reported by Golshan et al. [16], while the smallest one was presented by Lee et al. [12]. See Table 2.

**Reliability**—the largest gaps between patients’ expectations and perceptions were reported by Mohammadi et al. [13], while the smallest gap was presented by Lee et al. [12]. See Table 2.

**Responsiveness and assurance**—the largest gaps between patients’ expectations and perceptions were reported by Qolipour et al. [9] and the smallest gap was presented by Lee et al. [12]. See Table 2.

The mean results of expectation for the total group include tangibility—4.48 (SD = 0.53); reliability—4.57 (SD = 0.51); responsiveness—4.53 (SD = 0.52); assurance—4.57 (SD = 0.53); and empathy—4.52 (SD = 0.53) (see Table 2). The mean results of perception for the total group include tangibility—3.66 (SD = 0.0.67); reliability—3.75 (SD = 0.67); responsiveness—3.61 (SD = 0.76); assurance—3.77 (SD = 0.70); and empathy—3.78 (SD = 0.71) (see Table 2).

According to the random effect meta-analysis (I-squared index 99.0%, *p* < 0.001), the pooled estimate of MD was 0.82 (95% CI: 0.62–1.01), which indicated that the expectations of all the participants were significantly higher than their actual perceptions (*p* < 0.001) for tangibility dimension. The MD varied from 0.05 (Lee et al. [12]-patients) to 1.76 (Golshan et al. [16]-urolithiasis adherence patients) (see Figure 2). The funnel plot (Appendix A) and the results of Egger’s test indicate no evidence of publication bias in some of the studies (t = −0.23, *p* = 0.82).

According to random effect meta-analysis (I-squared index 99.3%, *p* < 0.001), the pooled estimate of MD was 0.82 (95% CI: 0.60–1.03), which indicated that the expectations of all the participants were significantly higher than their perceptions (*p* < 0.001) for the reliability dimension. The MD varied from 0.05 (Lee et al. [12]–patients; Naqavi et al. [5]) to 2.01 (Mohammadi et al. [13]) (see Figure 3). The funnel plot (Appendix A) and the results of Egger’s test indicate no evidence of publication bias in some of the studies (t = 1.34, *p* = 0.20).

According to a random effect meta-analysis (I-squared index 99.3%, *p* < 0.001), the pooled estimate of MD was 0.78 (95% CI: 0.55–1.01), which indicated that the expectations of all the participants were significantly higher than their perceptions (*p* < 0.001) in the responsiveness dimension. The MD varied from -0.03 (Lee et al. [12]–patients, (see Figure 4), the funnel plot (Appendix A) and the results of Egger’s test indicate no evidence of publication bias in some of the studies (t = 1.10, *p* = 0.29).

According to the random effect meta-analysis (I-squared index 99.0%, *p* < 0.001), the pooled estimate of MD was 0.80 (95% CI: 0.63–0.98), which indicated that the expectations of all the participants were significantly higher than their perceptions (*p* < 0.001) in the assurance dimension. The MD varied from 0.08 (Lee et al. [12]-patients) to 1.50 (Qolipour et al. [9]) (see Figure 5). The funnel plot (Appendix A) and the results of Egger’s test indicate no evidence of publication bias in some of the studies (t = 3.78, *p* = 0.01).

According to random effect meta-analysis (I-squared index 99.0%, *p* < 0.001) the pooled estimate of MD was 0.75 (95% CI: 0.57–0.92), which indicated that the expectations of all participants were significantly higher than their perceptions (*p* < 0.001) in the empathy dimension. The MD varied from −0.03 (Lee et al. [12]-patients) to 1.56 (Golshan et al. [16]-urolithiasis adherence patients) (see Figure 6). The funnel plot (Appendix A) and results of Egger’s test indicate evidence of publication bias in some of the studies (t = 2.95, *p* < 0.01).

The pooled estimated MD for tangibility and reliability was 0.82 (95% CI: 0.62, 1.01) and 0.82 (95% CI: 0.60, 1.03), respectively. The gap between the expectations and the perceptions was the lowest for empathy [0.75 (95% CI: 0.57, 0.92)] (see Figure 7). The funnel plot and the results of Egger’s test (Appendix A) indicate no evidence of publication bias in some studies (t = 0.58, *p* < 0.62).

## 4. Discussion

The quality of medical services is assessed by different tools and methods in particular countries. This quality can be approached in specific areas, such as reliability, responsiveness, tangibility, empathy, and assurance. The average patient’s expectations were shown as higher than the perceptions of service quality in all the dimensions, meaning that gaps did exist in all the above-mentioned areas. The study demonstrated that the largest gaps in the medical service quality assessments completed by patients were identified in the dimensions of tangibility and reliability, while the smallest ones in the areas of empathy and responsiveness (see Figure 7).

## 5. Tangibility

The material aspect or tangibility seems to be important for patients in hospital outpatient clinics, where an attractive outpatient environment and services provided at high levels are considered some of the most important reasons why the patients make use of hospital outpatient clinics [4]. Lee et al. showed that patients’ expectations in this domain were met relatively well (the smallest gap), however, the tangibility dimension is among the most serious of problems faced by Korean hospitals [12]. According to Al Fraihi et al., hospital managers should then improve the physical/technical conditions of the premises at which medical services are provides at outpatient clinics [4].

Naqavi et al. pointed out that the provision of healthcare outpatient units with modern medical equipment and devices being used efficiently would enable the service to be received in the shortest possible time and at a high quality level [5]. The same was also pointed out by patients in the study by Vafaee-Najar et al., who had had the highest expectations in relation to the material sphere [18]. Similarly, a large gap was identified in the tangible area (access to equipment) among patients with chronic kidney disease in the study conducted by Golshan et al. at an academic public institution [8,16]. Zun et al. pointed out that the physical appearance of premises and the lack of modern equipment at the GP clinic effectively reduced c patients’ satisfaction with the quality of received medical services [14]. At private medical facilities in Saudi Arabia, Iran, Jordan, and Malaysia, the tangible dimension, relating to the physical infrastructure, had the highest expectations among patients [20,21,22].

## 6. Reliability

In the area of reliability, Naqavi et al. noted that the delivery of medical services at an appointed time and the commitment of medical staff, as well as the maintenance of accurate medical records, significantly improved the quality of medical services [5]. Similarly, Mohammadi et al. identified the reliability gap as the most important one in the perception of patients and as a priority in improvement plans of healthcare facility managers [13]. Hercos et al. also stressed the view that special attention must be paid to reliability in the health care system [23]. Also, according to Goul et al., this domain is one of the most important for patients [24]. Reliability was also identified as the most important dimension in the study by Lee et al. [12] (patients). Nadi et al. carried out an analysis in a group of 600 patients admitted to internal medicine, surgery, and women’s and children’s wards at three different hospitals, indicating the largest gap between the actual state and the expectations of the patients in the area of reliability [6]. Other authors obtained similar results [17].

## 7. Responsiveness

Responsiveness refers to the levels of receptiveness, openness, sensitivity, and awareness of health professionals [4]. It also means the willingness of medical staff to assist patients, the presence/attendance of staff when asked/called by patients, and the adherence of doctors/nurses to service time schedules [19]. With regards to responsiveness, the patients’ expectations were fulfilled in Korea [12]. Golshan et al. demonstrated a failure to meet the expectations in patients who were not attending appointed follow-up visits and were not adhering to medical recommendations. Satisfaction with service quality was rated relatively low [16]. The reason for such an assessment of the service quality was related to short illness duration, i.e., the patient’s poor knowledge and awareness of his/her condition, medical staff unavailability, the time-consuming process of making follow-up appointments, the remoteness of a medical facility from the place of residence, and the patient’s failure to comply with the timing of follow-up appointments (numerous appointments related to the chronic treatment process) [16]. The domain of responsiveness was also rated low in a study by Qolipour et al., who found that providing prompt service to patients and setting an appointment for a test and ensuring that it is completed was an important part of the ‘responsiveness’ gap [9]. A study by Al Fraihi et al. demonstrated that patients of a hospital-based outpatient clinic highlighted the need for an easy system for appointment making and prompt telephone answering by the facility’s registrar [4]. Mohammadi et al. identified the responsiveness gap as one of the most important among the five gaps in the Servqual model [13]. The dimension of responsiveness was of moderate importance for patients in Bangladesh [15].

## 8. Assurance

The Qolipour et al.’s study showed that patients’ expectations were not met in the area of assurance. The authors reported that hospitals in Iran set up special service improvement committees to control clinical and medical quality [9]. Qolipour noted that, in order to increase the assurance of patients migrating to receive medical care (“medical tourism”), reputation, knowledge of doctors, and the expertise of the medical staff were really not enough. The quality of all the aspects of the healthcare services should be improved so that the patient feels really safe [9]. Research on medical service quality, carried out among medical tourism patients, identified negative gaps in all the analysed areas. The expectations of medical tourists (patients of orthopaedic, otolaryngological, obstetric, and gynaecological departments) were higher than their perceptions, so those patients were not satisfied with the standard of care received at the hospitals. That was true for both public and private hospital patients in Iran [9]. Golshan et al. showed that the individualised approach to the patient and the awareness and knowledge of doctors and staff of new medical techniques were reflected in the assessment of the assurance dimension among patients (best results). However, the identified discrepancy between expectations and perceptions in all the dimensions indicates a need to improve the quality of services at the hospital outpatient clinic of nephrology. The domain of assurance scored the best, while tangibility scored the weakest [16]. Golshan et al. also noted that the availability convenience of doctors’ hours scored the worst among adherent and non-adherent patients [16].

Fatima et. al. demonstrated that, in terms of assurance, patients’ expectations were largely met. Assurance was there related to the availability of competent, qualified, trained, and professional doctors, nurses, and other auxiliary staff [11]. In the study by Roy et. al., the reliability/diligence of hospital staff, including their availability was also of particular importance to patients [15]. According to Aghamolaei et al., the large discrepancy in the assurance gap was due to the poor communication of doctors, psychologists, nurses, and medical staff with patients. The quality of services at the University Hospital was at an average level, while the lowest rating was given to the efficiency and readiness of the staff to help, and the responsiveness domain was rated the worst [19].

## 9. Empathy

Empathy refers to the levels of understanding, sympathy, and compassion shown by medical staff [4]. In a study by Nadi et al., the authors indicated that empathy was most important, even though it did not demonstrate the largest gap between the actual status and the patients’ expectations [6]. Naqavi et al. points out that the understanding and respecting of the patient’s emotions and high quality interpersonal communication induce a greater sense of comfort and satisfaction with the medical service [5]. A study conducted at a hospital outpatient clinic showed that patients’ expectations in this dimension were met at a relatively high level, meaning that they were treated under complete privacy and with dignity, and that the medical staff were understanding their needs [4]. High expectations in this area were also presented by patients in studies by other authors [8,19]. A great gap in the dimension of empathy was also identified among medical tourists hospitalised in Iran [9]. According to Qolipour at al., in the domain of empathy, one should necessarily take into account dignity, confidentiality, autonomy, and communication [9]. Empathy expectations were not met in a study by Golshan et al., either, who listed the empathy dimension as one of the important elements of patient adherence to doctor’s recommendations [16]. In a study by Fatima et. al., empathy was rated high by patients at hospitals in Pakistan and was found to be an important aspect in the evaluation of the quality of medical services and patient care [11].

From the publications under analysis, it can be concluded that different patients were evaluated, but it was noted that, irrespective of the study site, elderly people, chronically ill people, and those with lower education levels were, in general, more critical regarding the quality of medical services [8,12,13,14]. In the studies of those populations, one may notice a greater gap between the status they expect and the status they find. Golshan et. al. demonstrated that the patients had high expectations in all the areas and perceptions of service quality, which may be due to cultural or sociodemographic characteristics [16]. Similar observations were made by other researchers who had singled out education level, low income, and the frequency of visits to medical centres among the important sociodemographic factors which may determine the patient’s satisfaction [14,25,26]. In addition, a significant relationship was found between the low education level of the patients and a higher satisfaction with the offered medical service [14,27,28]. It was also observed that people with higher levels of education were more critical and capable of being more objective in their perception of received medical services [14]. Analogous conclusions were drawn by Al Fraihi et al., namely that the quality gap in those patients was higher than in other patient groups, mainly in the dimensions of tangibility and assurance [4]. Qolipour et al. observed that, in order to increase the assurance of patient migrating for medical purposes, the professional level of medical staff was not enough. The quality of all the aspects of the healthcare services should be improved so that the patient feels really safe [9]. Not only those areas with the highest discrepancies between expectations and perceptions, since a gap in one dimension may exert a synergistic effect onto other dimensions of service quality, leading to an overall reduction in service quality [4]. In addition, on the basis of the conducted meta-analysis and the review of the literature, it seems that the problem of communication was one of the most frequently mentioned in the analysed papers [5,8,9,10,15,17,18,19]. The authors pay particular attention to the need for training in interpersonal and inter-group communication in order to meet the needs arising from the treatment process and to ensure the quality of this treatment. The results of other authors show that the service quality element of personal interactions and relationships is one of the most important components, affecting the patients’ perception of service quality [29,30,31]. In addition, their research shows that human factors have a greater impact on patients’ perceptions of quality than non-human aspects. Similar findings are presented by other authors, i.e., patients’ perceptions are of key importance for the general evaluation of service quality [32] and often concentrate on intangible quality, i.e., empathy, the sense of dignity and confidentiality [9], interpersonal communication [5,10,17,23,33], listening to the patient [8,33], the skills of doctors, hospital atmosphere, care and concern of the staff, involvement in the therapeutic process of patients [10], behaviour of staff during service delivery [11], the responsiveness to patients’ needs [19,23,33], and consideration of their opinions and comments in treatment planning [8].

## 10. Conclusions

The Servqual method is broadly used in various regions to assess the quality of medical services provided.The study demonstrated that patients had significantly higher expectations from offered medical services in the five analysed dimensions.The results, obtained through the Servqual method, may help improve the quality of services provided by different institutions.In the publications analysed, it was noted in all the cases that respondents had higher expectations of quality than the quality level they had received. In only two cases (not statistically significant) were the expectations comparable to the quality of service received. The difference was observed in South Korea and Saudi Arabia, where both countries demonstrate high standards of living. Patients most often make use of public health care institutions and want the services provided by them to be of high quality. However, high quality is not always ensured at every level at public health care entities. Gaps in one dimension may have an impact on the final quality of medical services, thus contributing to their under-rating by patients. In order to improve the quality of medical services, one should focus not only on the areas with the biggest gaps.Medical facilities should take measures to reduce differences in the quality of provided services. Differences in the quality of services should compel those responsible for management to take specific actions after any problems are identified. Depending on the country and facility surveyed, gaps in the quality of medical services provided were identified in different areas, primarily empathy/communication, medical equipment and facilities, and the waiting times for service delivery. The quality of healthcare services provided may improve the patients’ adherence to treatment and care recommendations [6]. On the basis of the analyses carried out, it is concluded that the problem of communication is one of the most frequently mentioned issues in the analysed works [7,8,9,10,11,12,13,14,15,16,17,18,25,26]. The authors pay particular attention to the need for training in interpersonal and intergroup communication in order to meet the needs arising from the treatment process and to ensure the quality of this treatment.

## Figures and Tables

**Figure 1 ijerph-19-07831-f001:**
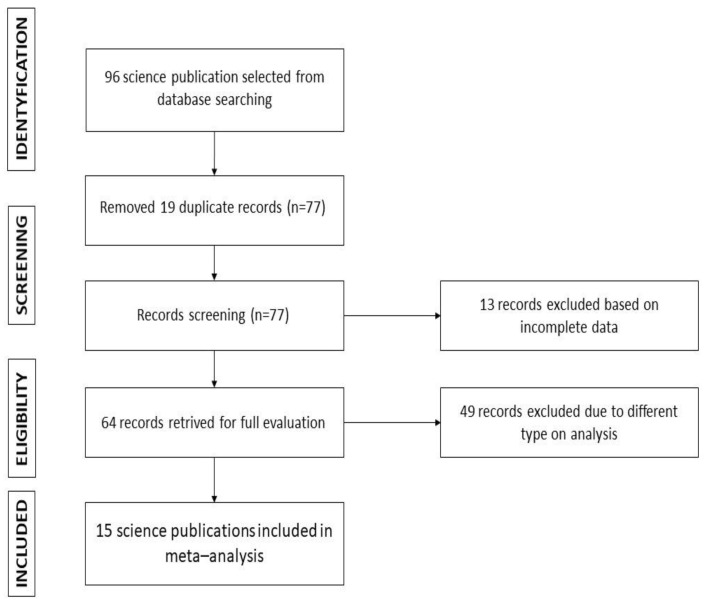
PRISMA flowchart describing the study design.

**Figure 2 ijerph-19-07831-f002:**
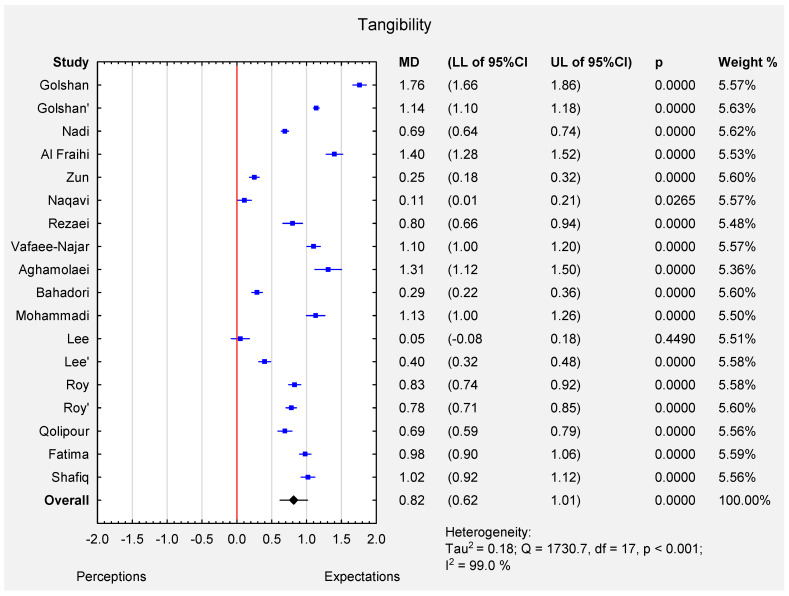
Forest plot for the mean difference ([MD]; 95% CI) of patients’ perceptions and expectations of tangibility dimension.

**Figure 3 ijerph-19-07831-f003:**
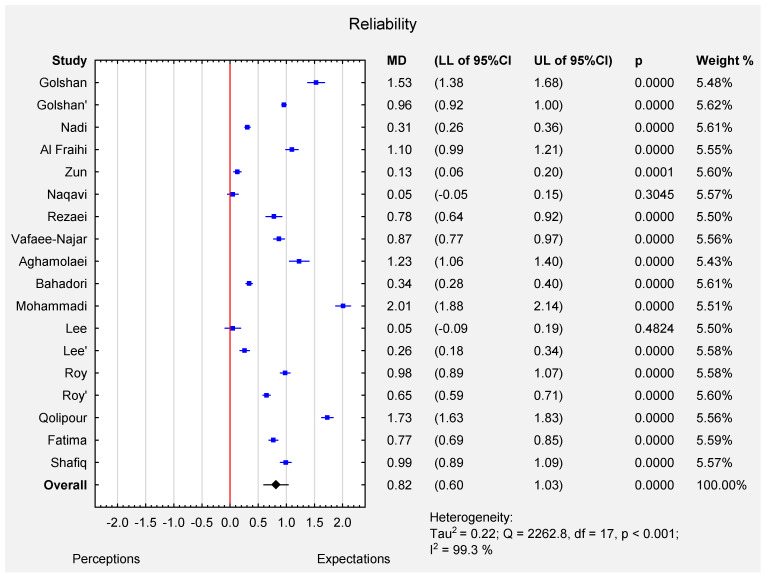
Forest plot for the mean difference ([MD]; 95% CI) of the patients’ perceptions and expectations in reliability dimension.

**Figure 4 ijerph-19-07831-f004:**
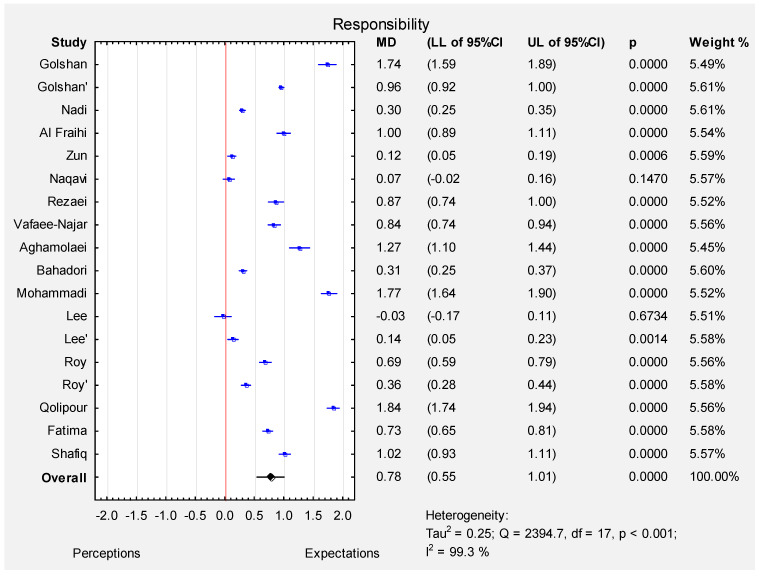
Forest plot for the mean difference ([MD]; 95% CI) of patients’ perceptions and expectations in the responsiveness dimension.

**Figure 5 ijerph-19-07831-f005:**
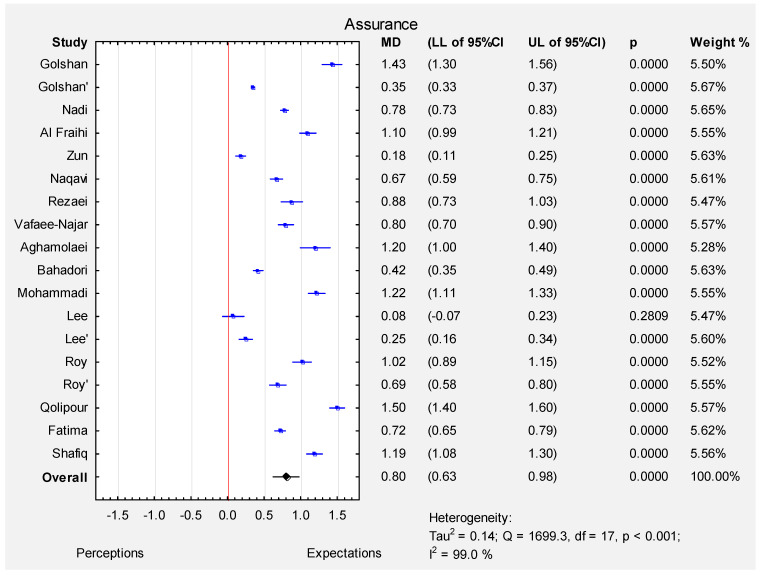
Forest plot for the mean difference ([MD]; 95% CI) of patients’ perceptions and expectations in the assurance dimension.

**Figure 6 ijerph-19-07831-f006:**
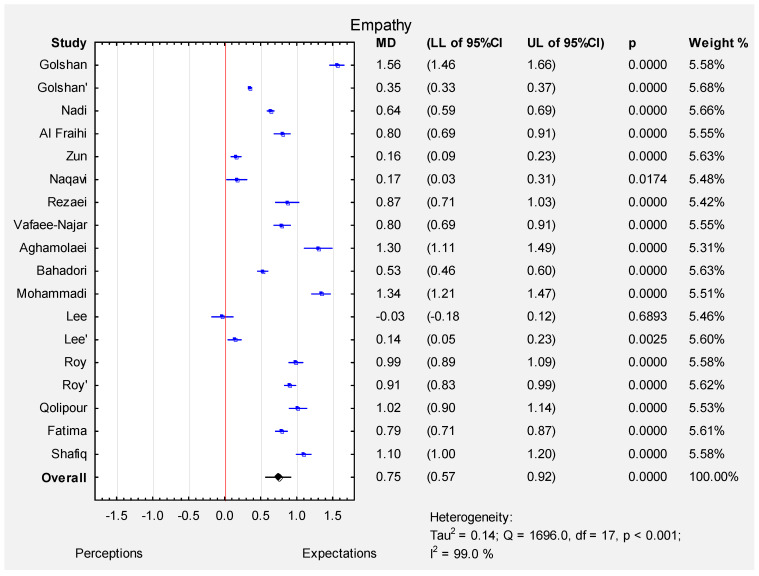
Forest plot for the mean difference ([MD]; 95% CI) of the patients’ perceptions and expectations in the empathy dimension.

**Figure 7 ijerph-19-07831-f007:**
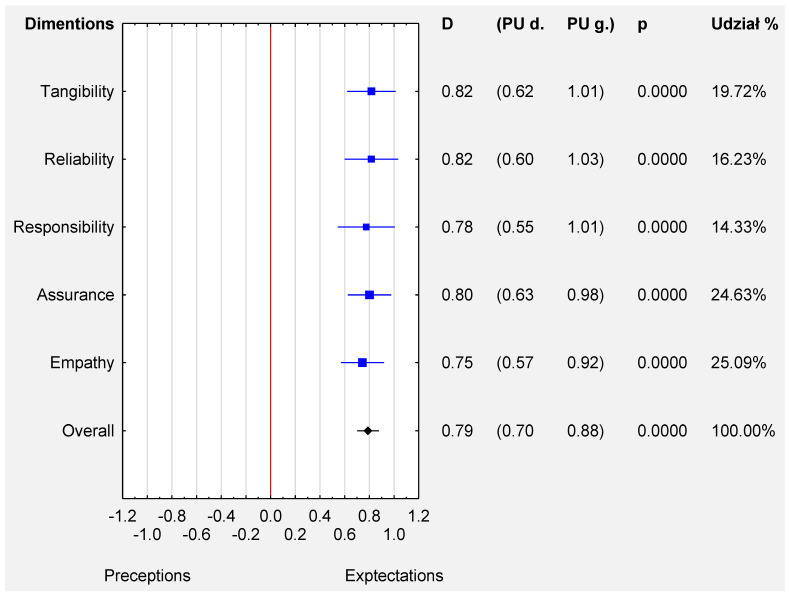
Forest plot for the mean difference ([MD]; 95% CI) of the patients’ perceptions and expectations in all dimensions.

**Table 1 ijerph-19-07831-t001:** Characteristics of the studies, included in this meta-analysis.

Author	Year	Country	Place of Research	No. of Participants (*n*)	Gender				Age [Years]		Marital Status				Comments
					Women		Men				Single		Married		
					*n*	%	*n*	%	Mean	SD	*n*	%	*n*	%	
Golshan [16]	2019	Iran	Hospital	51	20	39.2	31	60.8	44.7	4.39	109	20.5	422	79.5	Urolithiasis patients adherence
Golshan [16]	2019	Iran	Hospital	531	172	32.4	359	67.6	51.03	11.05	9	17.6	42	82.4	Urolithiasis patients non-adherence
Nadi [6]	2016	Iran	Hospital	600	439	73.2	161	26.8	39.94	10.99	64	10.7	536	89.3	The study population included patients with at least 24 h hospitalisation period at internal, surgical, gynaecological and paediatricsectors
Al Fraihi [4]	2016	Saudi Arabia	Hospital	306	117	38.2	189	61.8	NR	NR	NR	NR	NR	NR	Patients, patients in emergency condition
Zun [14]	2018	Malaysia	Public Clinic	386	202	52.3	184	47.7	37.3	13.5	NR	NR	NR	NR	Patients at the registration
Naqavi [5]	2014	Iran	Treatment centres	260	32	12.3	228	87.7	37	9.4	NR	NR	NR	NR	Drug addiction therapy
Rezaei [17]	2016	Iran	Hospital	400	148	37	252	63	38.5	23.6	NR	NR	NR	NR	Patients who were hospitalized for at least two days
Vafaee-Najar [18]	2014	Iran	Healthcare centre	435	387	89	48	11	30.30	10.37	33	7.58	401	92.18	Patients from physicians’ programmes
Aghamolaei [19]	2014	Iran	Hospital	89	32	36	57	64	32.9	10.05	NR	NR	NR	NR	Hospitalised patients
Bahadori [8]	2014	Iran	Haemodialysis centre	184	75	40.8	109	59.2	NR	NR	67	36.4	117	63.6	Chronic kidney disease
Mohammadi [13]	2010	Iran	Healthcare centre	300	300	100	0	0	28.4	NR	0	0	300	100	Patients of health care centres
Lee [12]	2006	South Korea	Hospital	272	121	44.5	151	52.6	NR	NR	NR	NR	NR	NR	Patients
Lee [12]	2006	South Korea	Hospital	282	282	100	0	0	NR	NR	NR	NR	NR	NR	Nurses
Roy [15]	2017	Bangladesh	Healthcare centre	200	90	45	110	55	NR	NR	NR	NR	NR	NR	Patients of public healthcare centres
Roy [15]	2017	Bangladesh	Healthcare centre	200	92	46	108	54	NR	NR	NR	NR	NR	NR	Patients of private healthcare centres
Qolipour [9]	2018	Iraq	Hospital	250	77	30.8	173	69.2	39.0	2.2	70	28	180	72	Iraqi tourists evaluating Iranian hospital services
Fatima [11]	2017	Pakistan	Hospital	817	423	51.8	394	48.2	37.4	16.6	NR	NR	NR	NR	Patients of Emergency, Surgical or Diagnostic Department
Shafiq [10]	2017	Pakistan	Hospital	340	NR	NR	NR	NR	NR	NR	NR	NR	NR	NR	Patients
TOTAL	-	-	-	5903	3009	54	2554	46	37.9	11.2	59	17	285	83	-

*n*—number of participants; %—percentage; NR = not reported.

**Table 2 ijerph-19-07831-t002:** The mean results of expectation and perception for the total group.

Author	Expectations										Perceptions									
	Tangibility		Reliability		Responsiveness		Assurance		Empathy		Tangibility		Reliability		Responsiveness		Assurance		Empathy	
	M	SD	M	SD	M	SD	M	SD	M	SD	M	SD	M	SD	M	SD	M	SD	M	SD
Golshan [16] (Urolithiasis patients adherence)	5.00	-	4.79	-	5.00	-	5.00	-	5.00	-	3.24	0.35	3.26	0.54	2.82	0.50	3.57	0.49	3.44	0.36
Golshan [16] (Urolithiasis patients non-adherence)	5.00	-	5.00	-	5.00	-	5.00	-	5.00	-	3.86	0.48	4.04	0.48	4.22	0.48	4.65	0.23	4.65	0.23
Nadi [6]	4.62	0.47	4.62	0.47	4.61	0.48	4.60	0.46	4.61	0.47	3.93	0.43	4.31	0.36	3.89	0.46	3.82	0.48	3.97	0.34
Al Fraihi [4]	4.60	0.56	4.60	0.54	4.50	0.60	4.60	0.54	4.70	0.53	3.20	0.90	3.50	0.82	3.20	0.83	3.50	0.83	3.90	0.84
Zun [14]	4.44	0.44	4.49	0.44	4.48	0.46	4.50	0.46	4.56	0.46	4.19	0.53	4.36	0.51	4.35	0.50	4.32	0.49	4.40	0.51
Naqavi [5]	4.26	0.55	4.40	0.54	4.42	0.53	4.45	0.43	4.27	0.90	4.15	0.58	4.35	0.57	4.27	0.60	3.78	0.53	4.10	0.72
Rezaei [17]	4.61	0.90	4.43	0.80	4.52	0.51	4.70	0.65	4.65	0.34	3.81	1.12	3.65	1.21	3.96	1.45	3.82	1.33	3.78	1.60
Vafaee-Najar [18]	4.53	0.56	4.48	0.57	4.45	0.65	4.48	0.62	4.27	0.71	3.43	0.86	3.61	0.88	3.39	0.95	3.68	0.90	3.47	0.97
Aghamolaei [19]	4.73	0.40	4.72	0.43	4.76	0.38	4.76	0.47	4.69	0.47	3.42	0.83	3.49	0.72	3.34	0.81	3.56	0.86	3.39	0.80
Bahadori [8]	4.30	0.35	4.60	0.22	4.57	0.29	4.72	0.27	4.37	0.35	4.01	0.38	4.26	0.32	4.21	0.35	4.30	0.36	3.84	0.34
Mohammadi [13]	4.33	0.76	4.56	0.66	4.32	0.63	4.28	0.72	4.17	0.71	3.20	0.89	2.55	0.98	2.49	0.96	3.06	0.70	2.83	0.89
Lee [12] (patients)	3.48	0.75	3.75	0.84	3.67	0.84	3.75	0.88	3.60	0.89	3.43	0.79	3.70	0.82	3.61	0.84	3.67	0.85	3.63	0.86
Lee [12] (nurses)	3.96	0.48	4.12	0.51	4.00	0.53	4.03	0.54	3.99	0.57	3.56	0.54	3.86	0.51	3.79	0.53	3.78	0.55	3.85	0.53
Roy [15] (Patients of public healthcare centres)	4.22	0.33	4.84	0.31	4.55	0.45	4.64	0.49	4.67	0.34	3.39	0.53	3.86	0.54	3.31	0.99	3.62	0.77	3.68	0.61
Roy [15] (Patients of private healthcare centres)	4.60	0.39	4.86	0.30	4.57	0.44	4.64	0.49	4.71	0.33	3.82	0.37	4.21	0.36	3.71	0.89	3.95	0.64	3.80	0.45
Qolipour [9]	4.61	0.41	4.73	0.39	4.84	0.35	4.76	0.38	4.82	0.33	3.92	0.70	3.00	0.73	2.94	0.74	3.26	0.73	3.80	0.92
Fatima [11]	4.66	0.64	4.65	0.66	4.61	0.73	4.68	0.59	4.65	0.67	3.68	1.00	3.88	0.96	3.88	0.99	3.96	0.89	3.86	0.99
Shafiq [10]	4.69	0.46	4.65	0.47	4.68	0.44	4.66	0.47	4.69	0.46	3.67	0.81	3.66	0.77	3.62	0.82	3.47	0.89	3.59	0.83
Total	4.48	0.53	4.57	0.51	4.53	0.52	4.57	0.53	4.52	0.53	3.66	0.67	3.75	0.67	3.61	0.76	3.77	0.70	3.78	0.71

M—mean; SD—standard deviation.

## Data Availability

Not applicable.

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
