# Peer review of "The Servqual Method as an Assessment Tool of the Quality of Medical Services in Selected Asian Countries"

_ijerph, 2022, doi:10.3390/ijerph19137831_

Round 1

Reviewer 1 Report

A meta-analysis was performed using the Servqual method in order to assess the quality of medical services provided. Now, my comments, which I hope they are useful for the authors.

Introduction

It is not clear why the study was only conducted in Asia, but worldwide. Thus, authors should mention in the introduction the rationale why the study was conducted in Asia and not in other parts of the world.

Results

Lines 133 to 141 are not clear when describing the dimensions. Those lines claim that tables 1 and 2 should be seen, but table 1 shows the characteristics of the included studies and it should only be table 2.

The studies included in this meta-analysis were: [4,5,6,8-12,15-20,28]. However, it is not clear why the Golshan studies are presented twice with different data. reference 18, Lee [12a] and Roy [17]. By the way, reference 12a is not presented, so it is understood that it refers to reference 12

The inclusion criterion is mentioned in the methods section “The number of studied patients over 90”. However, the following studies do not meet this criterion: First, they mention the study by Golshan [18] with 51 and that of Aghamolaei [28] with 89.

The title of figures 6 and 7 is repeated. Apparently, figure 7 seems to be wrong , since a comparison of the different dimensions is made and not only empathy. The description of Figure 7 is appropriate.

Discussion

The discussion seems more like a narrative description of the studies contained in the meta-analysis by dimension.

Authors should include the context of Asia, that is, this meta-analysis only focused on Asian countries, but these results were not contrasted with other studies carried out in other parts of the world, since they do not mention if studies of this type have been conducted  in America, Europe, Africa or Oceania.

The application of these results in Asia.

Conclusions

It is said that “1. The Servqual method is broad used in various regions to assess the quality of 358 medical services provided.”, however, it is not specified in which regions, that is, worldwide or in Asia, if it is in Asia. Please describe the location of the studies in those regions in the results section.

Author Response

Dear Review,

Thank you very much for your important comments. Thanks to your attention, our work has become better and brings even more to the topic of Servqual method. A table with responses to your comments is attached. Please see the attachment.

Best regards,

Dr Aleksandra Jonkisz 

Reviewer 2 Report

General Comments

Thank you for preparing this paper. It documents important findings. However, application of the JBI Institute framework for assessing systematic reviews and meta-analyses revealed two important omissions that must be addressed before consideration for publication.

While the Discussion is comprehensive, the current Conclusions section is  completely inadequate. Conclusions should succinctly summarize  key findings that will be of value to the readership in terms of shaping health service policy, health service facility planning, professional development etc. etc. 

Your Discussion is rich with examples that should be generalized to provide useful recommendations to policy makers, health service planners, clinician educators etc. etc. addressing the identified gaps between  pre-service use expectations and  post-use perceptions of the services as rendered.

The second omission relates to directives for new research, as this is one of the principal purposes of meta-analysis. These should be detailed at the end of the Conclusions section. 

Some Minor Edits

Line 34: Delete the word 'met'

Line 92: Remove the word 'studied' preceding the word 'patients'; replace the word 'over' with the symbol '>'

Line 294: Replace 'Acc.' with 'According' 

Line 311: Replace 'Ac.' with 'According' 

Appraisal Framework

The following is the completed framework developed by JBI International for reviewers of Systematic Reviews and Meta-analyses, for your reference.

1.         Is the review question clearly and explicitly stated?

□Y

2.         Were the inclusion criteria appropriate for the review question?

□Y

3.         Was the search strategy appropriate?

□Y

4.         Were the sources and resources used to search for studies adequate?

□Y

5.         Were the criteria for appraising studies appropriate?

□Y

6.         Was critical appraisal conducted by two or more reviewers independently?

□Y

7.         Were there methods to minimize errors in data extraction?

□Y

8.         Were the methods used to combine studies appropriate?

□Y

9.         Was the likelihood of publication bias assessed?

□Y

10.     Were recommendations for policy and/or practice supported by the reported data?

□N

11.     Were the specific directives for new research appropriate?

□N

https://jbi.global/critical-appraisal-tools 

Author Response

(The authors gave the same response as above.)

Round 2

Reviewer 1 Report

I strongly insist that authors should justify in the introduction the reasons why they chose Asia to conduct the study, as being from the Asian continent is one of the inclusion criteria. I suggest that it is their understanding that although there are studies that use the Servqual Method word wide, these studies do not have the appropriate methodological rigor in Asia, which explains why Asian countries are only included.

Author Response

Dear Reviewer,

Thank you very much for this comment. Our work has been improved. We kindly ask you to reconsider the publication of this work and we remain at your disposal.

Best regards,

Dr Aleksandra Jonkisz

Reviewer 2 Report

Thank you for your efforts in refining the Conclusions section. The paper is now complete in my view. 

Author Response

Dear Reviewer,

Thank you very much for your comments, they have significantly improved our work. We are grateful for this collaboration with you.

Best regards,

Dr Aleksandra Jonkisz